# Current Knowledge on the Pathophysiology of Lean/Normal-Weight Type 2 Diabetes

**DOI:** 10.3390/ijms24010658

**Published:** 2022-12-30

**Authors:** Teresa Salvatore, Raffaele Galiero, Alfredo Caturano, Luca Rinaldi, Livio Criscuolo, Anna Di Martino, Gaetana Albanese, Erica Vetrano, Christian Catalini, Celestino Sardu, Giovanni Docimo, Raffaele Marfella, Ferdinando Carlo Sasso

**Affiliations:** 1Department of Precision Medicine, University of Campania Luigi Vanvitelli, I–80138 Naples, Italy; 2Department of Advanced Medical and Surgical Sciences, University of Campania Luigi Vanvitelli, I–80138 Naples, Italy; 3Mediterrannea Cardiocentro, I–80122 Napoli, Italy

**Keywords:** type 2 diabetes mellitus, normal-weight, pathophysiology

## Abstract

Since early times, being overweight and obesity have been associated with impaired glucose metabolism and type 2 diabetes (T2D). Similarly, a less frequent adult-onset diabetes in low body mass index (BMI) people has been known for many decades. This form is mainly found in developing countries, whereby the largest increase in diabetes incidence is expected in coming years. The number of non-obese patients with T2D is also on the rise among non-white ethnic minorities living in high-income Western countries due to growing migratory flows. A great deal of energy has been spent on understanding the mechanisms that bind obesity to T2D. Conversely, the pathophysiologic features and factors driving the risk of T2D development in non-obese people are still much debated. To reduce the global burden of diabetes, we need to understand why not all obese people develop T2D and not all those with T2D are obese. Moreover, through both an effective prevention and the implementation of an individualized clinical management in all people with diabetes, it is hoped that this will help to reduce this global burden. The purpose of this review is to take stock of current knowledge about the pathophysiology of diabetes not associated to obesity and to highlight which aspects are worthy of future studies.

## 1. Introduction

The ongoing dramatic increase in incidence and prevalence of T2D is largely explained by the global epidemic of obesity, due to the known linkage between these two conditions [1]. It has been recently observed that any pattern of life-course exposure to obesity, i.e., thinness increasing weight or overweight from both early and late adulthood trajectories, may raise the risk for T2D development with respect to a stable normal BMI [2]. On the other hand, most overweight and obese individuals never develop diabetes, and conversely some normal-weight individuals are diabetic so that obese and non-obese subclasses of T2D are currently used in clinical practice.

An adult-onset non-insulin-dependent diabetes with low/normal BMI has been known since 1955, when Hugh-Jones first reported it in Jamaica as a unique form of diabetes among lean individuals that lacked the clinical features of both type 1 diabetes (T1D) and T2D [3]. Over the following decades, cases of leanness-associated diabetes emerged in many other low- and middle-income countries from Asiatic and African continents. In 1985, the WHO officially recognized a “malnutrition related diabetes mellitus” (MRDM) characterized by a resistance to the development of ketosis, partial resistance to insulin action, extreme degree of wasting and emaciation, and onset of symptoms before the age of 35 years [4]. In 1999, this clinical entity was dropped from WHO classification of diabetes due to a lack of substantial evidence for malnutrition or protein deficiency as independent causes of disease [5]. However, several epidemiologic data have continued to support the existence of lean diabetes in populations living in low-income countries of Asia and Africa, and others studies pointed to a high prevalence of diabetes in normal-weight non-white populations, even in high-income settings such as the U.S. [6,7,8,9,10]. More recently, some authors have classified the entire population of subjects with diabetes, both T1D and T2D, into five clusters based on the presence of six variables (age at diagnosis, BMI, glutamate decarboxylase antibodies, HbA_1c_, β-cell function, and insulin resistance). Those subgroups seem differently associated with diabetes complications and the classification could help clinicians identify patients at higher risk and tailor therapy [11].

Nevertheless, the lean/normal-weight T2D remains, to date, an understudied topic that deserves consideration for at least two reasons. Firstly, the largest increase in diabetes prevalence in the coming years is expected in non-obese, non-white individuals who represent the majority of the world if considering that nearly half of the globally estimated 463 million adults with diabetes live in India and China [12]. Due to the continuous migratory flows of Asian and other ethnic groups such as asylum-seekers refugees from conflict areas in the Middle East and Northern Africa, the number of people with diabetes will dramatically increase even among non-white ethnic minorities living in high-income Western countries [13,14]. A second point for interest is the still dilemmatic “obesity paradox”. Decade ago, a great sensation was raised by the results of a pooled analysis of longitudinal observational studies showing that adults with normal weight at the time of diabetes appearance had higher mortality than those overweight or obese [15]. Similar results were reported in other studies on individuals with diabetes [16,17]. In recent large meta-analyses, U-shaped associations between BMI and all-cause mortality in people with diabetes were observed with nadirs in the range of overweight or mild obesity [18,19,20]. Other authors did not confirm the phenomenon, or remarked the poor appropriateness of BMI as a measure of adiposity [21]. Moreover, they observed that most studies did not control for preexisting chronic illness and smoking status, or suffered from other limitations such as reverse causation, selection and treatment bias, and a one-time measurement of weight [22]. Even taking into account these opposite data, a protective effect of excess weight on mortality in diabetes cannot be denied with certainty and the question remains unsolved [23].

Whereas scientists from all over the world have spent great efforts to understand the mechanisms linking obesity to diabetes, the pathophysiology of lean T2D is still debated. Knowing eventual specific metabolic and physiologic drivers of the disease in normal-weight individuals is of great relevance for implementation of an effective prevention and an individualized clinical management in order to reduce the global burden of diabetes [24].

The aim of this review is to discuss the state of the art knowledge about the pathophysiology of diabetes not associated to excess weight and to highlight what are the peculiarities compared to the more common obese diabetes. Pathophysiological characteristics of lean diabetes are synthesized in Figure 1.

## 2. Epidemiology of Lean/Normal-Weight T2D

A few studies have assessed the frequency of T2D in white normal-weight populations, for example those documenting a prevalence of 5.1% in Australian adults and an incidence of 15% in a male Swedish cohort [25,26].

According to data collected in the last twenty years, populations from Asia and Africa are at risk for T2D at much lower levels of BMI than other ethnic groups, suggesting the need for lowering their current targets for ideal body weight [27,28].

A 2.1% incidence of MDRD was reported in a sub-Saharan African rural area, and a study examining the T2D prevalence in Zambia and Western Cape of South Africa found values of 2.9% and 9.4%, respectively, two-thirds of which were associated to under- or normal-weight [8,29].

In a nationally representative sample from China, T2D prevalence was 4.5% in individuals with a BMI < 18.5 kg/m^2^ and 7.6% in those with a BMI of 18.5–24.9 kg/m^2^ [7]. A similar 7.8% prevalence was found in subjects with a BMI < 25 kg/m^2^ in a nationally representative survey from mainland China [30]. In an underdeveloped area of South China, 68.2% of instances of newly detected T2D were from non-obese diabetics [31]. In a Japanese population, over 60% of the subjects with diabetes were not obese [32]. 

India is referred as one of the T2D capitals, with a predicted prevalence of 151.4 million indigenous South Asians affected by 2045 [33]. A study in semi-urban/rural India found most hyperglycemia in undernourished people [6]. Overall, T2D in Asian Indians is characterized by younger age of onset and relatively low BMI [34]. An analysis of the U.K. Biobank containing four large ethnic groups, established that to have the same diabetes risk as white participants with a BMI  > 30 kg/m^2^, the equivalent BMI in South Asians was only 22 kg/m^2^ [35]. Cross-sectional analyses using representative samples of Asian Indians (South Asia CARRS-Chennai Study) and whites (U.S. NHANES Survey) demonstrated a significant ethnic difference in T2D prevalence in men, 5.4%/23.5% in under- and normal-weight Asian Indians and 0.0%/6.1% in their white counterparts. In women, the prevalence was 5.6%/13.6% in under- and normal-weight Asian Indians and 2.3%/2.8% in whites [36].

Most data come from ethnic minorities living in high-income Western countries. A study cohort of 18,000 T2D patients living in Chicago showed that around 13% had a BMI ranging from 17 to 25 kg/m^2^ and that Asians had a five-fold higher prevalence in the lean group (17% vs. 4%) [9]. In a large racially/ethnically and geographically diverse cohort of adults who were all members of integrated health care systems to correct for disparities in access to services, the age-standardized prevalence of diabetes increased across BMI categories among all groups. However, the prevalence of diabetes and prediabetes at low to normal BMI was 3.5 and 5%, respectively, in white subjects, and 7.3 to 10.2% and 9.6 to 18% in the various racial/ethnic groups, being the higher values registered in Hawaiians/Pacific Islanders and Asians [10]. A recently published systematic review and meta-analysis of prospective cohort studies (minimum 12-month follow-up, over 2.69 million participants from 20 countries) using ethnic-specific BMI categories, emphasized the crucial role of obesity demonstrating an increasing T2D risk of 0.93 for underweight, 2.24 for overweight, 4.56 for obesity, and 22.97 for severe obesity with respect to normal-weight. Interestingly, the underweight resulted a protective factor against T2D only in non-Asian people (RR = 0.68, 95% CI: 0.40–0.99, I^2^ = 56.1%, *n* = 6) [37].

Overall, these results suggest that environmental and genetic factors beyond obesity may contribute to the disproportionate burden of disease in non-white populations with ancestry from low- and middle-income countries.

## 3. Pathophysiology of T2D Development in Absence of Excess Weight

In concert with phenotypic variations, T2D presents a heterogeneous pathophysiology resulting from the variable interplay between insulin resistance and β-cell dysfunction in dependence of demographics, genetics, and clinical characteristics of patients [38].

According to the more accepted model performed in obese subjects of European descent, under conditions of obesity-related insulin resistance, β-cells are stimulated to secrete more insulin than in normal insulin sensitivity. Once secretory function cannot cope with the defective hormonal action, hyperglycemia and eventually T2D develop, with the contribution of concurrent glucolipotoxicity and inflammation [39,40,41,42].

In underweight/normal-weight categories, insulin secretion deficiency and insulin resistance may have a respectively higher and lower burden than in obese with diabetes, and an early failure of β-cells may result in appearance of diabetes at much lower insulin resistance [39]. In this direction, a lower basal and early-phase insulin secretion and a lower degree of insulin resistance have been documented in young T2D Japanese individuals with a BMI of 23.4 kg/m^2^ than in whites with diabetes and a BMI of 27.5 kg/m^2^ [43,44].

### 3.1. Role of β-Cell Failure

The key feature of non-obese T2D seems to be a defect in the insulin secretion capacity as opposed to peripheral insulin resistance described in classical diabetes.

Dated studies in patients with MRDM and BMI < 18 kg/m^2^, reported fasting C-peptide levels intermediate between those of T1D and T2D subjects, and both basal and stimulated insulin plasma levels consistently lower than obese with diabetes, though enough to avoid ketosis [45,46,47,48,49].

In the Chicago study, a rapid β-cell failure is suggested as a major pathophysiologic mechanism by both higher prevalence and early initiation of insulin use in lean individuals [9]. In recent investigations using homeostatic model assessment of β-cell function (HOMA-β) and of insulin resistance (HOMA-IR), non-obese Filipino T2D patients presented lower β-cell function and lower insulin resistance with respect to their overweight/obese counterparts [50]. Also, sub-Saharan Africans with incident adult-onset non-autoimmune diabetes at BMI < 25 kg/m^2^ predominantly exhibited β-cell secretory dysfunction rather than insulin resistance [51]. A cross-sectional analysis of two community-based cohorts living in the U.S. showed a significantly higher age-adjusted prevalence of diabetes in the South Asians of MASALA study (23%) than in ethnic groups of Multi-Ethnic Study of Atherosclerosis (MESA)(6% in whites, 18% in African Americans, 17% in Latinos, and 13% in Chinese Americans). The data were likely explained by the lower HOMA-β and the higher HOMA-IR exhibited by South Asians after adjustment for age and adiposity [52]. Prospective findings from the British Whitehall study best characterized the pathogenesis of T2D in South Asian individuals. Changes in glucose levels, insulin sensitivity and insulin secretion observed for several years before diabetes diagnosis, emphasized important differences in the natural history of T2D in adults of European (mean BMI = 26.3 kg/m^2^) with respect to those of South Asian (mean BMI = 24.3 kg/m^2^) ethnic origin. These presented a much earlier impairment of β-cell function leading to rapidly increasing fasting plasma glucose and diabetes development. Overall trajectory data in South Asians were suggestive for a long-term β-cell compensation elicited by a chronic insulin resistance detectable from childhood, and for an inability to produce further insulin to overcome the decreasing insulin sensitivity after the 60 years [53].

A recent cross-sectional study found that the mean age of T2D onset in two cohorts of non-migrant Asian Indians was dramatically lower than in a cohort of white Europeans from Scotland (47–50 vs. 62 years), and that diabetes onset at age ~40 years was more frequent at a normal ethnicity-specific BMI (prevalence, 24–39% vs. 9.3%) [54]. Interestingly, Asian Indians with normal BMI and young onset of diabetes had lower values of HOMA-β compared with those diagnosed at an older age and those diagnosed young with overweight or obesity [54].

Applying to a population of Indians with young-onset T2D the diabetes stratification in five clusters, recently identified in newly diagnosed unselected European patients, the cluster of severe insulin-deficient diabetes (SIDD) predominated; whereas that of mild obesity-related diabetes (MOD) was the most prevalent in young Europeans [11,55,56]. These patterns confirmed that the deficient insulin secretion rather than the insulin resistance, often called into play, was the driver of young-onset T2D in India, in contrast to obesity and insulin resistance in young Swedish and Finnish T2D populations.

### 3.2. Role of Insulin Resistance

Previous investigations described insulin resistance as an intrinsic defect of non-insulin-dependent diabetes, irrespective of obesity, being the deleterious effect of diabetes on in vivo insulin-stimulated glucose utilization much greater than that of obesity [57,58,59]. On the other hand, insulin resistance was suggested by biochemical alterations observed in muscle biopsies of lean without diabetes first-degree offspring of T2D parents [60]. Especially insulin resistance characterizes non-diabetic normal-weight individuals displaying obesity-related phenotypic features whereby it has been associated with activation of the JNK pathway in muscle biopsies [61,62].

These findings bring us back to the role of fat accumulation and distribution in the genesis of insulin resistance even in non-obese T2D. A study addressing the relationship between fat distribution and insulin resistance, found that for each level of total and regional adiposity, non-insulin-dependent subjects with diabetes had a heightened state of insulin resistance than control subjects without diabetes and with similar degrees of obesity [63]. Similarly, Taniguchi et al. showed in non-obese Japanese T2D patients an independent association of insulin resistance with both subcutaneous and visceral fat areas [64]. Instead, a study in black Americans with the same type of diabetes demonstrated that visceral but not subcutaneous abdominal fat volume was associated with insulin resistance [65].

The metabolic phenotype of Asian Indians, defined “thin-fat” and markedly different from white people, is characterized by increased truncal fat present at birth and through childhood and adult life, despite a normal or low BMI [54]. Moreover, even if insulin deficiency seems to represent the main driver for T2D in these people, their insulin resistance appears more marked than in European subjects. Indeed, Asian Indians show a diabetes risk at BMI of 22 kg/m^2^ which overlaps that of white people at BMI > 30 kg/m^2^ [35,66,67]. A study in non-obese T2D Asian Indians considering separately lean (BMI < 19 kg/m^2^) and non-lean non-obese (BMI > 19 to <25 kg/m^2^) individuals, showed a similar β-cell function in the two groups. However, higher surrogate markers of insulin resistance were associated to higher intra- and total abdominal adiposity in non-lean non-obese diabetics [68]. Insulin resistance and truncal obesity have been described to account for the excess incidence of diabetes in African Caribbean ethnic minority [67].

## 4. Factors Influencing Β-Cell Function and Insulin Sensitivity in Non-Obese Diabetes

### 4.1. Distribution of Body Fat

The literature suggests a positive and linear relationship between BMI and T2D risk [37]. Sadly, measurement of BMI to classify body size phenotypes provide a biased estimation of body composition, since no discrimination between lean and fat body mass, and within this last, between subcutaneous and visceral fat. Of importance, BMI gives no information about ectopic lipid deposition in liver, muscle, heart, pancreas, and vasculature, all organs with great implication in the pathophysiology of diabetes and cardio metabolic risk [69,70]. So true is it that some obese individuals called metabolically healthy obese (MHO), scarcely present obesity-associated pathologies, whereas others defined as metabolically obese normal weight (MONW) exhibit obesity features despite having normal weight [71,72]. This second category could include the non-obese patients with diabetes.

According to the “adipose expandability” hypothesis, the storage capacity of subcutaneous adipose tissue is limited and intrinsically different in each individual. Crossed the limit, the risk increases for redirecting lipid accumulation toward other adipose depots and/or non-lipocyte cells. The consequent lipotoxic insults including insulin resistance and inflammation, are the critical players in the development of adipose-related metabolic diseases and T2D [73,74,75,76,77]. Genetically dictated mechanisms, as well as environmental factors (i.e., early malnutrition), would determine the ability for subcutaneous adipose expansion and the predisposition to visceral adiposity [78,79,80].

Moreover, as suggested by some authors, adipose tissue expansion is shown through adipocyte hyperplasia, hypertrophy, or a combination of both. Those histological properties seem relevant to determine its metabolic effects. More specifically, hypertrophy of adipose tissue, both subcutaneous and visceral, seems to be associated with a pernicious metabolic profile. On the other hand, subcutaneous adipose tissue hyperplasia seems to be associated with a better metabolic phenotypic profile [81,82]. As recently observed, adipose hypertrophy is associated with a low generation rate of adipocytes, whereas hypoplasia is associated with high generation rates [83].

#### 4.1.1. Visceral Fat and T2D Incidence in Non-Obese Subjects

The role of body fat distribution in risk stratification has been poorly investigated in individuals without obesity and high-quality data on measurements of fat mass in cohorts with non-white participants are scarce.

Two longitudinal studies clearly documented visceral fat as a stronger predictor of T2D and metabolic syndrome than BMI, but they do not produce evidence of this association in the specific setting of non-obese population. In multiethnic individuals with serial adiposity assessments by computed tomography (CT) from MESA study, visceral but not subcutaneous fat, both at initial time point and over time changing, were strongly associated with incident metabolic syndrome, regardless of weight, race, age, or sex [84]. Kuwahara et al. investigated the BMI trajectory patterns before diabetes diagnosis and examined the associated changes in visceral adiposity and glucose metabolism in about 24,000 Japanese participants without diabetes [85]. They showed that those not developing diabetes had a relative steadiness of all measured variables during observation. Among people becoming diabetic, all BMI categories (low, medium, and high) showed an absolute or relative rise in visceral fat and an impaired β-cell compensation for increasing HOMA-IR, suggesting the worsening insulin resistance as the mediator of visceral fat leading to T2D [85]. In a just published study, Mongraw-Chaffin et al. specifically addressed in 1005 non-obese (BMI = 18.5 to < 30.0 kg/m^2^) African Americans of Jackson Heart Study, the longitudinal association of body fat distribution at CT scan with development of glucose metabolism disturbances [86]. They found that higher visceral fat, subcutaneous fat, BMI, and insulin resistance (HOMA-IR) were significantly associated with higher odds of incident metabolic syndrome and T2D with a strong effect for visceral fat. Interestingly, the association of visceral fat with metabolic syndrome persisted in normal-weight participants (BMI: 18.5 to <24.9 kg/m^2^) disturbances [86].

#### 4.1.2. Fatty Liver Disease and Diabetes Risk in Non-Obese People

Non-alcoholic fatty liver disease (NAFLD) is typically linked to obesity but can also develop in a substantial proportion of individuals with normal or even low BMI, a condition termed “lean NAFLD” frequently overlooked in clinical practice [87,88,89,90]. The disease is associated with a doubled incidence of new-onset diabetes, and may confer a diabetes risk even in normal-weight or under-weight individuals [71,91,92,93].

A recent study observed a higher prevalence of NAFLD diagnosed by CT in non-obese individuals with diabetes than in a non-obese control group, with an incidence of severe forms nearly seven times higher in those with T2D [94]. Interestingly, in a large longitudinal study in non-obese euglycemic adults, those with NAFLD had a risk of incident diabetes similar to that of overweight/obese people, emphasizing the involvement of liver fat in the early pathogenesis of T2D even in non-obese people [95].

A study in non-obese Asian Indians with T2D indicated that NAFLD and non-alcoholic fatty pancreas disease, could contribute to respectively insulin resistance and progressive β-cell dysfunction [96].

Racial differences in amounts of visceral adipose tissue may explain why some groups are at higher cardiometabolic risk even in the lower BMI categories [97,98,99]. The most demonstrative case is that of Asian Indians who have, at equal BMI, a higher body fat percentage, a prominent abdominal obesity, a higher intramyocellular lipid, and/or a higher liver fat content compared to Caucasians [100].

### 4.2. Reduced Mass of Skeletal Muscles

As suggested by the 2018 consensus of the writing group for the European working group on sarcopenia in older people 2 (EWGSOP2), sarcopenia is a muscle disorder, defined by both the reduction in the muscle mass and strength, and by an impairment of quality of muscle [101].

The so-called “sarcopenic obesity”, a different condition affecting lean but metabolically obese patients having both an excess of adiposity and a deficit in muscle mass, may promote the development of leanness-associated T2D and explain the “obesity paradox” [102,103].

Considering that skeletal muscle is the most insulin-sensitive tissue that accounts for 70–90% of post-prandial glucose disposal, a lower muscular mass relative to adiposity may be a plausible contributor to T2D risk [104]. However, whereas the relationship of excess adiposity with T2D is well recognized, the role of fat free mass related to T2D development remains less clear.

Data from the U.S. National Health and Nutrition Examination Survey III (NHANES III) demonstrated that a lower ratio of skeletal muscle mass (estimated by bioelectrical impedance) to total body weight in individuals under 60 years was strongly associated with both insulin resistance and dysglycemia, suggesting sarcopenia as an early predictor of diabetes susceptibility [105,106]. Other findings from NHANES III confirmed the association of a lower ratio between appendicular lean mass and BMI with greater insulin resistance in elderly patients [107]. In a population of U.S. adults, the percent lean mass was negatively associated with the values of glycosylated hemoglobin in both men and women without diabetes [108].

Some studies highlighted an inverse association of lean mass with incident T2D, whereas others did not [109,110,111,112,113]. Particularly, some authors highlight the different metabolic role of muscle and fat and suggest the fat to muscle ratio (FMR) as a predictor of higher risk of developing T2DM [114]. FMR is obtained from bioelectrical impedance and calculated by dividing the fat mass by the muscle mass [115]. In this 11-year prospective cohort study with UK Biobank data enrolling almost 500,000 subjects, high regional and total FMR was associated with a higher risk of developing T2DM. This result was independent of BMI and waist circumference. Recent findings are also contradictory. In a prospective cohort study conducted in a largely white population in Denmark, there was no association between lean body mass and incident T2D [116]. In a Finnish study investigating the combined effect of lean and fat mass index derived from bioelectrical impedance analysis, a greater muscle mass was not protective whereas a high lean mass associated to a high fat mass predicted subsequent development of T2D [117]. Finally, in a population from 2005–2006 NHANES, a reduced skeletal muscle mass independent of android and gynoid adiposity, was associated with higher diabetes prevalence in young men but not in women [118]. This sex discrepancy agrees with previous sex-stratified cross-sectional studies reporting that a relatively lower muscle mass was associated with insulin resistance and diabetes in men but not in women [111,119].

Apart from sex, many aspects of the relationship between muscle mass and T2D need clarification, such as the role of age and race disparities.

### 4.3. Genetic Architecture of Non-Obese T2D

As revealed by genome-wide association (GWA) studies, T2D is a complex polygenic disease with large pathophysiologic insights of genetic architecture both in obese and non-obese patients [120]. In these lasts, the genetic factor mainly controls the β-cell function and the body fat distribution.

#### 4.3.1. Genetic Susceptibility to β-Cell Dysfunction

Although the natural history of T2D is described traditionally as a process of increasing insulin resistance followed by progressive insufficiency of insulin secretion, a primary genetic impairment of β-cell function has been hypothesized. On this basis, even a little insulin resistance due to obesity, age, or other factors could precipitate the T2D development [121]. This way of thinking is likely the most fitting to non-obese T2D, where β-cell functional exhaustion is the major pathophysiologic mechanism compared to the prevalent insulin resistance in obesity with diabetes.

GWA studies in Europeans found that many variants with the strongest effects on diabetes were associated with reduced β-cell function, and that most (~80%) of these loci were more associated with lean T2D, implying a prevalent genetic predisposition in this phenotype [122,123]. At the other extreme, obese subjects presumably need fewer risk variants to develop diabetes since their predisposition due to weight excess and insulin resistance.

Perry et al. identified the association with T2D of LAMA1 gene encoding laminin-1 and involved in insulin secretion in lean Europeans, and Okamoto et al. identified the KCNJ15 gene as causative of β-cell dysfunction in T2D Asian patients [123,124].

Even if most common genetic variants are similar across populations of South Asian, East Asian, and European ancestry, suggesting a shared genetic susceptibility to T2D, other alleles vary in frequency across ethnicities and partly explain some features of diabetes among Indian and Chinese populations [125]. A large GWA meta-analysis conducted for T2D in East Asians identified eight loci reaching genome-wide significance, mostly related to β-cell functions. This genetic structure involves a more fragile β-cell resulting in early destruction and apoptosis, likely accounting for the high rates of diabetes at lower BMIs in this population [126]. Large GWA studies have confirmed that some genetic loci, such as PAX4 (encoding a pancreatic islet transcription factor), PSMD6 and ZFAND3 (encoding proteins involved in insulin secretion), NID2 (associated with lipodystrophy traits or body fat distribution), and ALDH2 (encoding an enzyme for alcohol metabolism associated with T2D in men), are unusually frequent among East Asian people [125]. In this ancestry, the polymorphisms of PAX4 and PSMD6 loci, which appear to be monomorphic in people from not East Asia, might associate to a reduction in insulin secretion and β-cell functional mass [127]. In a recent study using partitioned polygenic risk score for T2D, a valuable tool to understand patterns of disease predisposition and etiology, Asian Indians showed a higher genetic burden of risk variants for lower β-cell function. This issue confirms the decline in insulin secretion as the primary driver of young-onset normal-weight T2D in India [54].

According to the results of a GWA study, the genetic architecture of T2D in sub-Saharan Africans appeared characterized by several risk loci shared with non-African ancestral populations. Among these, the gene encoding the transcription factor seven-like (TCF7L2) known to affect pancreatic secretory function was the most significant [128]. Recently, a novel genomic locus (ZRANB3) associated to increased β-cell apoptosis, was identified in this population [129].

#### 4.3.2. Genetics of Insulin Resistance and Adipose Tissue Distribution

Genetic studies of obesity and body fat distribution estimate the heritability of BMI to be approximately 40–70% based on familial aggregation analysis, and of visceral and subcutaneous adipose tissue measured by CT scans, to be 36% and 57%, respectively [130].

By using data from large-scale GWA studies, Yaghootkar et al. observed that the 11 common genetic variants associated with insulin resistance clustered with a subtle “lipodystrophy-like” phenotype, conferring an increased risk for T2D and metabolic syndrome despite relative leanness [79]. These results suggested that a genetically directed higher visceral-to-subcutaneous fat ratio may be the foundation for the “metabolically obese, normal-weight” phenotype.

Various genetic data support a role of ethnic differences in body fat distribution for the diabetes risk. A study among Chinese Han individuals identified significant associations of genetic variants in or near CDKAL1, CDKN2BAS, and KCNQ1 with the risk for lean T2D, and of genetic variants near KCNQ1 or in FTO with the risk for obese T2D. In the same population, lean T2D patients had a stronger genetic predisposition for T2D risk alleles than obese counterparts, and the T2D genotype risk score for T2D contributed to a greater β-cell dysfunction [131]. In the U.K. Biobank, people of south Asian origin had a lower frequency of genetic variants associated to subcutaneous adiposity compared to white population, which might predispose them to increased visceral adiposity and adipocyte hypertrophy, resulting in insulin resistance and β-cell glucolipotoxicity [132].

Other investigators aimed to investigate the role of genetic variants for insulin secretion and resistance in T2D risk among normal-weight, overweight, and obese individuals. They found that whereas the insulin secretion score had a stronger association with T2D in leaner individuals, the role of polygenic insulin resistance was independent of body size [80]. The association of variants for insulin resistance even with lower subcutaneous fat mass suggested impaired adipose expandability and ectopic fat deposition among lean individuals [80].

Recent preliminary findings suggested that the encoding gene of chemerin, a chemoattractant protein involved in the pathogenesis of metabolic syndrome, may be a significant predictor of insulin resistance in a population of non-obese patients with T2D [133].

### 4.4. Early Life Malnutrition

Clinical and experimental evidence have demonstrated the vital role of under- and over-nutrition during early life in programming metabolic disorders such as obesity and T2D in adulthood [134].

Poor nutrition, especially protein deficiency, during fetal development and early childhood may drive lean T2D in adult life, as suggested by its higher frequency in underdeveloped and developing countries and by observational studies in people experiencing famine in worldwide regions [135]. Increasing popularity of vegetarian and vegan diets could become a novel risk factor due to inadequate amount and bioavailability of proteins from plant sources [136,137]. The causative relationship between early-life protein deficiency and later-life lean T2D has been well documented in a rat model where a gestational low-protein programming produced a progressively worsening T2D with the lean phenotype [138].

Among the mechanisms modifying growth and metabolic set points of fetuses living in a low protein environment, the epigenetic underpinning of glucose homeostasis is the most accredited [139]. All types of modifications may be involved, i.e., methylation in promoter region, acetylation of histone proteins, and regulation of miRNAs.

Epigenetic changes in the promoter region of phosphoenolpyruvate carboxykinase involved in the metabolic pathway of gluconeogenesis, induce impairment of glucose homeostasis in liver of rats born from a protein-restricted pregnancy [140]. Maternal low-protein diets altered the epigenetic control of the transcription factor Hnf4a gene implicated in T2D etiology, in rat pancreatic islets, and that of the glucose-6-phosphatase. The latter is a crucial enzyme in gluconeogenesis and glycogenolysis in newborn piglet livers [141,142]. In low-protein in utero programmed rats, modifications of histone in the promoter region of the insulin-2 gene decreased the number of pancreatic transcripts, and that of histone code impaired the gene expression of glucose transporter 4 [143,144]. In newborn piglets, protein-restricted pregnancy caused sex-dependent epigenetic alterations of the mitochondrial DNA-encoded OXPHOS activity which is involved in energy homeostasis by controlling electron transfer and ATP generation [145]. An increased expression of miR-15b reduced both β-cell mass and insulin levels in mouse offspring of maternal protein restriction [146]. Elevated expression of miR-615, miR-124, miR-376b, and decreased expression of miR-708 and miR-879 in maternal low-protein programmed mice modulated the offspring metabolic health from the weaning age [147].

It has been observed that the epigenetic modifications induced by undernutrition did not reverse after two generations of unrestricted rats, implying that any change in in utero nutritional status may cause permanent alterations in the fetal gene expressions [140,144]. The same can be extrapolated to multigenerational maternal undernutrition in human populations, as suggested by the characteristic Indian ‘thin-fat’ phenotype and the growing epidemic of diabetes in India [148].

The effects of low-protein fetal programming are mostly documented in pancreatic islets, but may involve all other organs engaged in glucose metabolism. Gestational protein restriction in rats caused insulin resistance and defective oxidative respiration in skeletal muscle, one of the main sites for peripheral glucose disposal [149,150,151]. Functional and structural changes of the liver associated to malnutrition during pregnancy may account for insulin resistance and impaired suppression of hepatic glucose production [152,153]. Alterations of hypothalamic nuclei notoriously involved in the central nervous regulation of food intake, body weight, and metabolism are described in weanling offspring of rat dams with low-protein regimens during gestation and lactation [154].

In addition to epigenetic changes, low birth weight and catch-up growth during early life, due to maternal malnutrition, increase the risk for chronic non-communicable diseases such as T2D [155]. In this connection, nesfatin-1 resistance in the hypothalamic arcuate nucleus of non-obese protein-restricted rats has been recently described to induce intrauterine growth retardation, thus contributing to the development of T2D [156]. In Japanese adults without diabetes, birth weight and β-cell mass were positively correlated [157].

### 4.5. Influence of Sex

A male preponderance for lean T2D up to 60% was reported in the U.S. ethnic minorities, Asian Indians, and sub-Saharan Africans [9,51,158].

Although the exact causes of this sex-related inequality are not clearly understood, a series of factors may be considered.

Genetic mechanisms support a greater tendency to accumulate abdominal fat for men and gluteo-femoral fat for women, being android fat more associated with insulin resistance and T2D than gynoid fat [159,160]. A study in sheep showed that males were more susceptible for visceral adiposity and obesity-related diseases than females, and in a MESA study, individuals with above-median visceral adiposity were more frequently male in each BMI category [84,161].

Since sex-specific differences in the expression of molecular markers of fat tissue differentiation and/or function emerge in utero, some authors hypothesized an adverse implication of late gestation and early postnatal malnutrition [162,163]. Investigations denied a relation with early nutrition history, but demonstrated that the subcutaneous fat had sex-specific upper-limits for expandability markedly lower in males than females [164].

Sex hormones and sex-dependent expression of genes may represent other factors determining the male preponderance of lean T2D and many studies indicated that women have better insulin sensitivity than men [165,166,167]. This phenomena could be related to the protective role of estrogen through stimulation of the mitochondrial biogenesis in white adipocytes or to inhibition of the hepatic transcription factor Foxo1 via activation of ERα-PI3K-Akt signaling in hepatocytes [168,169].

Acquired insults might be implicated if considering that some studies described a higher prevalence in males than females of smoking and alcoholism among lean T2D patients [9,170]. Chronic alcohol consumption may significantly contribute to the β-cells weakening [171]. In a 10-year longitudinal study in adult Koreans with baseline BMI < 23 kg/m^2^, alcohol consumption of at least 2 units/day significantly increased the appearance of T2D compared with lifetime abstainers [172]. Instead, exposure to passive and active smoking has been positively and independently associated with a risk of diabetes in women [173].

### 4.6. Islet Cells Auto-Antibodies and Metabolome Profiling

In a study among youth onset (<30 years) individuals with diabetes from North India, the co-occurrence of auto-antibodies against glutamic acid decarboxylase 65 (anti-GAD65) and tyrosine phosphatase-like protein (anti-IA-2) was found in 4.7% of subjects with ketosis-resistant lean diabetes and 22.4% of patients with T1D. Instead, anti-GAD65 positivity alone was seen in 38% of ketosis-resistant subjects with respect to the 14.2% in T1D subjects [174]. Another study in T2D Asian Indians found the presence of anti-GAD65 in 25.3% of T2D subjects, being the positive group younger and with lower HOMA-β than negative one [175]. It must be pointed out that any single antibody by itself is neither sensitive nor specific to distinguish between T1D and lean T2D since a low-titer GAD65 autoimmunity has been reported even among obese T2D people [176]. On the other hand, a meta-analysis associated the anti-GAD65 positivity with an increased risk for future T2D in adult people [177]. The role of autoimmunity in lean subjects with diabetes needs further elucidation to unearth whether these antibodies are involved in the β-cell destruction or their occurrence is merely secondary.

Various studies have characterized an obesity-related plasma metabolome including amino acids, metabolites of amino acid catabolism, lipids, and nucleotides that highlights a potential metabolic dysregulation. Several of these metabolites have shown BMI-independent associations with future risk of T2D [178]. In 7663 individuals from three population-based cohorts, lean individuals with an obesity-related metabolome had an increased risk for T2D compared with lean individuals having a healthy metabolome. Particularly, middle-aged normal-weight individuals with a metabolite fingerprint of BMI (mBMI) > 5 kg/m^2^ above their actual BMI, had a doubled risk of future T2D when compared with individuals who were normal-weight according to both BMI and mBMI [179].

### 4.7. Role of Microbiota and Adipokines

There is growing evidence concerning the influence of gut microbiome on the pathophysiology of metabolic disorders frequently associated with each other, including obesity, insulin resistance, and T2D [180]. The topic was little investigated in non-obese T2D. A recent study based on metagenomic sequencing data from 182 lean and abdominally obese individuals with and without newly diagnosed T2D, showed a depletion of *A. muciniphila* abundance in the gut microbiota of lean participants with T2D, associate with a reduction in secretion of insulin and expression of fibroblast growth factor 19 (FGF19), a protein that functions as a hormone regulating bile acid synthesis and glucose and lipid metabolism [181]. In the same study, supplementation with *A. muciniphila* protected mice against diet-induced glucose intolerance. On these bases, authors proposed a beneficial impact for this gut microbiota member leading to reduced blood glucose levels and improved glucose homeostasis. Little later, other investigators described a lower abundance of *A. muciniphila* in T2D patients with high HOMA-IR and BMI. However, this negative correlation was found in T2D patients with high lean tissue but not in those with high fat tissue, thus emphasizing the importance of body composition in the relationship between gut microbiota and insulin resistance [182].

Disturbances in adipocytokine secretion by adipose tissue contributing to both insulin resistance and impairment of insulin production are frequently described in metabolic diseases such as obesity and T2D [183,184]. Kocot et al. observed a significant increase in leptin and visfatin levels and decrease in adiponectin concentration in a population of obese T2D patients in comparison with healthy controls [185]. Similarly, Liu et al. demonstrated higher serum concentrations of leptin and lower serum concentrations of adiponectin in obese individuals with newly diagnosed T2D than their counterparts with normal BMI [186]. A recent small study conducted in Saudi Arabia concluded that significantly altered concentrations of adipokines were found in T2DM patients compared to controls, with more pronounced alterations in obese and highly obese individuals [187]. On the whole, these data indicate an association of altered adopokine profile with obesity rather that with diabetes per se, likely as a key factor in the inflammatory mechanisms of obesity. Conversely, the literature does not report specific alterations of adipokine levels in non-obese T2D [188].

## 5. Concluding Remarks and Future Directions

The multifaceted nature of T2D and its enormous complexity may be schematized in a varying combination of four main pathogenic abnormalities involving insulin secretion and sensitivity, and body fat amount and distribution, each with a non-uniform relative contribution among patients. The etiology of these alterations lies in a number of environmental risk factors associated to a combination of multiple, potentially thousands, of common etiological genetic variants and multiple low-frequency and rare genetic variants, not all identified in the GWA studies. Based on the data collected so far, the only distinctive feature among the aforementioned pathogenic factors for non-obese T2D is the normal/low body fat. Its distribution is possibly altered, as well as β-cells function and insulin sensitivity, generally—to a greater extent the first and less the second—with respect to the much more common T2D associated to obesity. In our opinion, this might be not enough to classify non-obese T2D as an independent entity with its own precise peculiarities, but rather just a variant positioned at low extreme of the phenotypic spectrum based on BMI. On the other hand, the great variety of names attributed over time to lean diabetes (Jamaica-type diabetes, MRDM, metabolically obese normal weight, malnutrition-related diabetes mellitus, etc.) expresses the lack of unmistakable intrinsic features within a great pathophysiologic complexity as classically observed in T2D.

There is still work to be done. Primarily, carefully designed mechanistic studies, including genetic and epigenetic investigations, are warranted without neglecting the possibility of not-yet explored pathways. These studies could deepen the underlying pathophysiology of non-obese T2D by longitudinal assessments of the relative contributions and timing of impaired insulin secretion and insulin resistance.

It must be further elucidated that the involvement of other individuals and environmental risk determinants, and in particular that of ectopic fat accumulation and lean mass. In this connection, it is not known whether the weight loss recommended for all overweight or obese with diabetes could be also effective in reducing ectopic fat accumulation in lean subjects or if it is rather harmful contributing to sarcopenia.

The amount of body fat is the phenotypic feature of T2D most taken in account by clinicians for its immediacy. When in excess, it induces automatically to consider an important presence of insulin resistance. Instead, in subjects classified as lean or underweight by BMI and particularly in males, there may be a high risk to underestimate the development of metabolic derangements induced by visceral fat even in absence of obesity. The reason why, a risk stratification by detection of body composition through non-traditional tools, such as dual-energy X-ray absorptiometry, is fundamental not to miss a prevention opportunity for lean subjects. In this setting, trials on diabetes prevention in non-overweight but high-risk individuals are of importance, given the increasing global prevalence of this diabetes phenotype and its lower age of onset determining a great impact on development of chronic complications. Since the earliness of detection, diagnosis, and treatment are important components of T2D prevention and health care, thus we must start from the assumption that preventive interventions needed across the full range of BMI. To ease the task, it might be useful to develop screening tools based on simple clinical parameters that can identify among non-obese individuals those at risk for developing T2D.

Pathogenic heterogeneity of T2D is rarely considered when deciding the class of antidiabetic agents in clinical practice and data on the most appropriate treatment strategies in non-overweight subjects with diabetes are scarce. In a time of increasing diffusion of personalized medicine, longitudinal studies specifically designed to assess the effectiveness of treatments promoting the preservation and recovery of β-cell function and that of insulin sensitivity in non-obese individuals with diabetes should be encouraged.

## Figures and Tables

**Figure 1 ijms-24-00658-f001:**
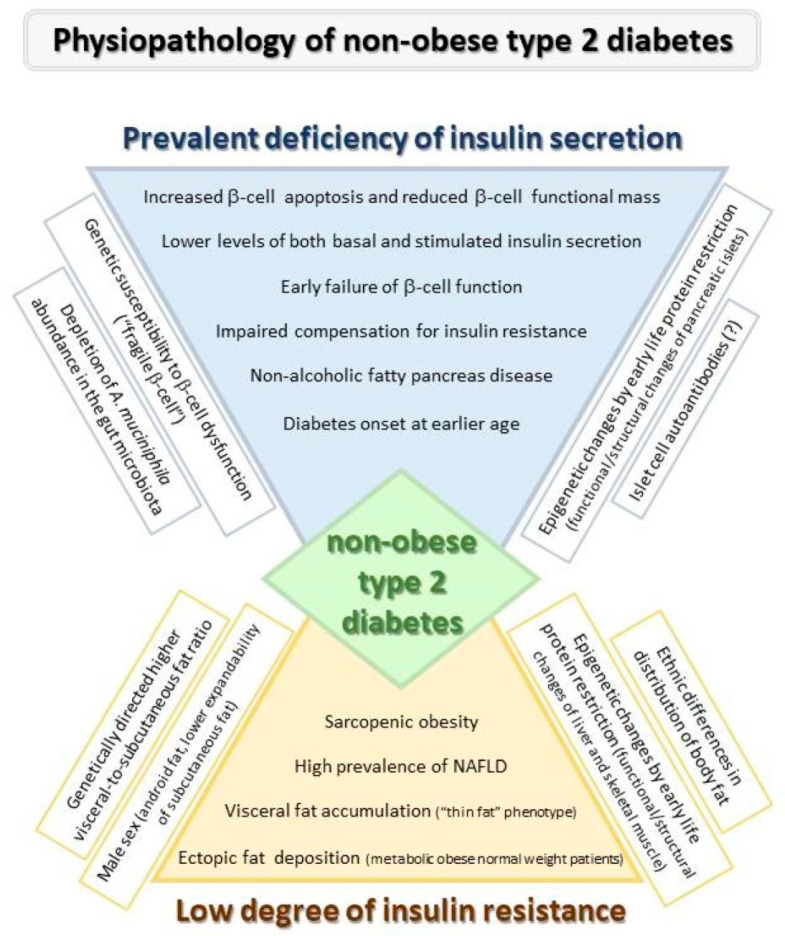
Pathophysiological characteristics of lean Diabetes.

## Data Availability

Not applicable.

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
