# Peer review of "Current Knowledge on the Pathophysiology of Lean/Normal-Weight Type 2 Diabetes"

_ijms, 2022, doi:10.3390/ijms24010658_

Round 1

Reviewer 1 Report

Savaltore T et al propose a review that focuses on the current knowledge on the physiopathology of type 2 diabetes (T2D) in normal-weight persons. This subject is not often treated in a review and an update of the different factors that could be involved in non-obese diabetic patients is of importance. This review is well written and designed but some minor modifications could be made in order to improve its quality.

1) In paragraph 4.1 concerning the distribution of fat, it will be also important to add a sentence and reference on the notion of hypertrophic versus hyperplasic adipocytes associated to “adipose expandibility” since hypertrophy has been shown to be a factor that has adverse effects on adipose tissue development compared to hyperplasia.

2) For a non-initiate reader, sarcopenia (also sarcopenic obesity) should be better defined since it concerns both a deficit in muscle mass and strength as reported by the European working group (EWGSOP2) in Age and Ageing, Vol 48 (1), 2019, 16–31. Moreover, some studies reported the importance of the fat to muscle ratio (FMR) as an indicator to assess T2D risk where fat mass was measured. This could be discussed as a complement to the cited papers (NHANES III study) considering total body weight. 

3) Among the different factors, the role of epigenetic and microbiota have not been treated. Could also others hormones (different from sex hormones) play a role? For example, some adipokines often associated to insulin resistance. It will be interesting to know if these different factors have been studied non-obese diabetic patients. Will it be possible to discuss it or to include them in the last part of the review (concluding remarks and future directions)? 

4) What could be a good complement to this review is a figure or a schematic drawing that summarizes the main particularities associated to T2D in persons without obesity. It will give a visual “take home message”.

5) page 8 line 360 « decliine in insulin secretion » is written instead of decline.  

Author Response

Salvatore T et al propose a review that focuses on the current knowledge on the physiopathology of type 2 diabetes (T2D) in normal-weight persons. This subject is not often treated in a review and an update of the different factors that could be involved in non-obese diabetic patients is of importance. This review is well written and designed but some minor modifications could be made in order to improve its quality.

Re: We wish to thank the reviewer for his/her precious comments, which helped us to improve the draft. All the suggested modifications of the draft are marked in yellow.

1) In paragraph 4.1 concerning the distribution of fat, it will be also important to add a sentence and reference on the notion of hypertrophic versus hyperplasic adipocytes associated to “adipose expandibility” since hypertrophy has been shown to be a factor that has adverse effects on adipose tissue development compared to hyperplasia.

Re: Accordingly, we briefly discussed this point and added some sentences and references. Line 251-258

2) For a non-initiate reader, sarcopenia (also sarcopenic obesity) should be better defined since it concerns both a deficit in muscle mass and strength as reported by the European working group (EWGSOP2) in Age and Ageing, Vol 48 (1), 2019, 16–31. Moreover, some studies reported the importance of the fat to muscle ratio (FMR) as an indicator to assess T2D risk where fat mass was measured. This could be discussed as a complement to the cited papers (NHANES III study) considering total body weight. 

Re: Accordingly, we revised the definition of sarcopenia and discussed the role of the fat to muscle ratio (FMR), by adding references.

3) Among the different factors, the role of epigenetic and microbiota have not been treated. Could also others hormones (different from sex hormones) play a role? For example, some adipokines often associated to insulin resistance. It will be interesting to know if these different factors have been studied non-obese diabetic patients. Will it be possible to discuss it or to include them in the last part of the review (concluding remarks and future directions)? 

Re: A number of investigations have focused on the epigenetic basis of obesity and obesity-associated T2D. The few data of literature involving epigenetic changes in lean T2D refer to the relationship of this type of diabetes with perinatal exposition to undernutrition that have been already described in the paragraph 4.4. (Early life malnutrition), lines 467-481. In the “Concluding remarks and future directions” the usefulness of epigenetic studies has been already mentioned (line 589,590). A paragraph has been added concerning role of microbiota and adipokines in the pathophysiology of non-obese T2D (paragraph 4.7).

4) What could be a good complement to this review is a figure or a schematic drawing that summarizes the main particularities associated to T2D in persons without obesity. It will give a visual “take home message”.

Re: Accordingly, we added a figure.

5) page 8 line 360 « decliine in insulin secretion » is written instead of decline.  

Re: Accordingly, we modified the text. Line 390

Reviewer 2 Report

The review by Salvatore et al. is a well-written piece that highlights in my opinion the many problems of the existing diabetes classification system.

I only have some minor comments:

1.     Abstract and throughout the text: please avoid using the term 'diabetic' (eg diabetic patients) as it is considered a stigmatizing language. The terms "people with diabetes" or "individuals with diabetes" are more appropriate. 

2.     Lines 122-124: Were these differences statistically significant?

3.     I would suggest adding a figure presenting the main pathophysiological characteristics of lean diabetes.

4.     It is worth discussing the Swedish study by Ahlqvist et al. (https://doi.org/10.1016/s2213-8587(18)30051-2) showing that a classification system based on risk of complications could be more effective than the classical division of people with diabetes into T1 and T2.

Author Response

The review by Salvatore et al. is a well-written piece that highlights in my opinion the many problems of the existing diabetes classification system.

I only have some minor comments:

Re: We wish to thank the reviewer for his/her precious comments, which helped us to improve the draft. All the suggested modifications of the draft are marked in yellow.

  1. Abstract and throughout the text: please avoid using the term 'diabetic'(eg. diabetic patients) as it is considered a stigmatizing language. The terms "people with diabetes" or "individuals with diabetes" are more appropriate.

Re: Accordingly, we modified Abstract and the text.

  1. Lines 122-124: Were these differences statistically significant?

Re: For what concerns this reference, the p value was not specified. However, we inserted the full result concerning this point. Line 136

  1. I would suggest adding a figure presenting the main pathophysiological characteristics of lean diabetes.

Re: Accordingly, we added a figure.

  1. It is worth discussing the Swedish study by Ahlqvist et al. (https://doi.org/10.1016/s2213-8587(18)30051-2) showing that a classification system based on risk of complications could be more effective than the classical division of people with diabetes into T1 and T2.

Re: Accordingly, we discussed that classification system based on risk complications, by adding the suggested reference. Line 56-61